# Provenance detection through learning transformation-resilient watermarking

## Abstract

Advancements in deep generative models have made it possible to synthesize images, videos and audio signals that are hard to distinguish from natural signals, creating opportunities for potential abuse of these capabilities. This motivates the problem of tracking the provenance of signals, i.e., being able to determine the original source of a signal. Watermarking the signal at the time of signal creation is a potential solution, but current techniques are brittle and watermark detection mechanisms can easily be bypassed by doing some post-processing (cropping images, shifting pitch in the audio etc.). In this paper, we introduce **ReSWAT (Resilient Signal Watermarking via Adversarial Training)**, a framework for learning transformation-resilient watermark detectors that are able to detect a watermark even after a signal has been through several post-processing transformations. Our detection method can be applied to domains with continuous data representations such as images, videos or sound signals. Experiments on watermarking image and audio signals show that our method can reliably detect the provenance of a synthetic signal, even if the signal has been through several post-processing transformations, and improve upon related work in this setting. Furthermore, we show that for specific kinds of transformations (perturbations bounded in the $\ell_2$ norm), we can even get formal guarantees on the ability of our model to detect the watermark. We provide qualitative examples of watermarked image and audio samples in the anonymous code submission link.

## 1 Introduction

Generative models have contributed to impressive advancements in content generation and representation learning in both digital image and audio domains (Brock et al. (2018); Kalchbrenner et al. (2018); Mehri et al. (2016); Zhu et al. (2017); Prenger et al. (2019); Donahue & Simonyan (2019); Oord et al. (2016); Goodfellow et al. (2014); Kingma & Welling (2013)). However, as generative models learn to better match a target distribution, the distinction between natural signals and synthetic signals generated by a model has blurred, leading to a raft of concerns over the potential misuse of these models (Chesney & Citron (2018)). For example, synthetic videos that are indistinguishable from natural videos, sometimes referred to as *deep fakes*, have the potential to cause widespread distrust in traditional media.

While the problem of detecting synthetic signals is interesting in its own right, it is challenging to do in a manner that is independent of the model used to generate the signal. Instead, we consider the problem of provenance detection via watermarking, a technique that involves injection of a carefully chosen (but imperceptible) perturbation (watermark) into the signal at the time of creation. The presence of the perturbation in the signal can be later used to detect the provenance (or ultimate source) of this signal. While the simplicity of the technique makes watermarking an appealing technique for detecting synthetic (as well as naturally generated) signals, it is susceptible to adversarial actors that can systematically try to remove the watermark by transforming the signal (so as to obfuscate the provenance). In the case of images, this may take the form of cropping, addition of Gaussian noise or blurring, rotating images etc. Many watermarking schemes break down under these types of transformations.

In this paper, we propose a novel *transformation-resilient watermarking* scheme- Resilient Signal Watermarking via Adversarial Training (ReSWAT ) that is able to detect the presence of a watermark

even after the signal has been through systematic attempts to remove the watermark. We use ideas from the literature on adversarial training (Madry et al., 2017) to learn to synthesize watermarks (encoding the provenance of a signal) that can be detected even after the watermarked signal has been through transformations that seek to remove the watermark. Our detection method can operate in any domain with continuous data representations. Experimental results demonstrate that our method is effective and substantially improves over existing methods - it can reliably detect the provenance of a synthetic signal even if the signal has been deformed or manipulated.

## 1.1 CONTRIBUTIONS

1. We formulate the transformation resilient watermarking problem and show how it can be reduced to an empirical risk minimization problem with a minimax loss function.

2. We develop a transformation resilient watermarking scheme, named ReSWAT (Resilient Signal Watermarking via Adversarial Training) and show that the learning procedure is closely related to adversarial training techniques (Madry et al., 2017; Athalye et al., 2017).

3. Empirically, we show across a set of image and audio datasets that our scheme can produce imperceptible watermarks that can be detected even after the signal has been through a series of adversarial transformations that preserve fidelity to the original signal. We do this using both standard metrics for signal watermarking and via human evaluation on audio signals. Furthermore, we formal provable guarantees of detection if an adversarial transformation has a bounded $\ell_2$ norm.

## 1.2 RELATED WORK

The problem of detecting synthetic signals has recently received a significant amount of attention. Marra et al. (2018) investigate how a generative adversarial network (GAN) (Goodfellow et al., 2014) leaves an identifiable fingerprint in the image it generates, while Nataraj et al. (2019) detect if an image was generated by a GAN by training a detection model on co-occurence matrices of the RGB channels of images. Similarly, Yu et al. (2018) also train a model that learns to detect if an image was generated by a GAN, and attribute a generated image to its source model. Unfortunately, these detection methods suffer from some limitations: (1) they are specifically designed to operate only in the image domain and (2) they are not designed to be resilient to transformations that an attacker could apply.

Digital watermarking consists of a two stage process: An embedding stage, where the original signal is combined with a hidden message (also referred to as the *watermark*), producing a watermarked version of the signal. Secondly, a *decoding* stage, where the hidden message is retrieved from the watermarked signal. Watermarking schemes are typically evaluated on fidelity of the watermarked signal to the original signal, the resilience of the watermark to various post-processing transformations and the false positive rates (i.e. whether the decoder detects watermarks in non-watermarked signals). Robustness of watermarking techniques is measured against a number of practical attacks that aim to remove or degrade the watermark signal. For example, adding Gaussian noise, image cropping and image compression represent watermark degrading, watermark removal and watermark geometric (re-positioning) attacks, respectively. In this paper, we are primarily interested in zero-bit watermarking, where one is simply interested in detecting the presence of a watermark (rather than decoding a hidden message from the watermark). The state-of-the-art zero-bit watermarking scheme in the image domain is referred to as *Broken Arrows* (BA) (Furon & Bas, 2008) and won the "Break Our Watermarking System" (BOWS) competition (Bennour et al., 2007). BA has a provable minimum false positive rate under Gaussian noise, however the scheme does not provide robustness against geometric attacks (like rotations and cropping).

## 2 FORMULATION OF THE ROBUST WATERMARKING PROBLEM

We study signals that live in a space $\mathcal{X}$ and are generated by a probabilistic source $P_s$ - our framework applies regardless of whether $P_s$ is a natural source (images of natural scenes) or an artificial source (samples from a VAE or a GAN). We are interested in developing a scheme to watermark signals generated by this source, with the requirement that:

1 The detector should detect a watermark in any watermarked signal from source $P_s$ and not detect a watermark in any other signal.

2 Even if post-processing transformations (for example, compression/cropping/frequency shift etc) are applied to the signal, the detector detects the watermark, or the absence thereof.

Formally, we define the robust watermarking problem as follows:

**Definition 2.1** (Transformation resilient watermarking scheme). Consider a watermarking scheme $(W, D)$ where $W : \mathcal{X} \mapsto \mathcal{X}$ is a watermarking routine and $D : \mathcal{X} \mapsto \{0, 1\}$ is a watermark detector. Let $\mathcal{T}$ be a space of transformations with $T : \mathcal{X} \mapsto \mathcal{X}$ for each $T \in \mathcal{T}$. We consider the watermarking scheme resilient with respect to a set of transformations $\mathcal{T}$ for a source $P_s$ if $D\big(T\big(W(s)\big)\big) = 1, D\big(T(s)\big) = 0, \forall T \in \mathcal{T}$ with high probability for $s \sim P_s$.

This formulation suggests an empirical risk minimization approach for training $D$. Given a fixed $W$, we can train $D$ to minimize the empirical risk

$$\underset{s \sim P_s}{\mathrm{E}} \left[ \max_{T \in \mathcal{T}} \ell\Big( D\big(T\big(W(s)\big)\big), 1 \Big) + \ell\Big( D\big(T(s)\big), 0 \Big) \right]$$

where $\ell$ is a loss function measuring the discrepancy between the prediction of $D$ and the desired label (0 for non-watermarked signals and 1 for watermarked signals).

The inner maximization transformations resembles the objective used in adversarial training (Madry et al., 2017), motivating our approach to learning a transformation resilient watermarking. We develop this idea in the following section.

## 3 ReSWAT : Resilient watermarking via adversarial training

We parameterize the detector $D$ as a neural network $f_\theta$, parameterized by $\theta$. If we fix the detector, the watermark embedding process $W$ should embed a watermark that is "strongly detected" by the detector, i.e., the watermark embedding mechanism pushes the watermark deep inside the decision boundary of $D$. However, we also want the watermark to be imperceptible so we require that the watermark does not significantly change the original signal according to some distance measure. While it is challenging to define the space of imperceptible distortions of the input, a convenient proxy is to limit the change in terms of the $\ell_\infty$ norm between the watermarked and the original signal. Thus, we construct the watermark, $\delta$, by solving the following optimization problem:

$$W(s) = \min_{\|\delta\|_\infty \leq \epsilon} \ell(f_\theta(s + \delta), 1)$$

This optimization problem can be solved efficiently using a projected gradient descent (PGD) method, and is mathematically very similar to computing an adversarial example for a neural network (however, in this case, the adversary is friendly and actually pushes the example to minimize the loss of the detector). The parameter $\epsilon$ controls the perceptibility of the watermark. If $\epsilon$ is too large, the watermark would be clearly perceptible while if it is too small, the watermark may easily be washed out by post processing steps. Thus, the right choice of $\epsilon$ achieves the optimal trade-off between perceptibility and transformation resilient detection. Note that we use the $\ell_\infty$ norm of the perturbation instead of another norm since this distributes the watermark over the entire signal, and so will be resilient to geometric attacks (that black out patches of the input, or obfuscate inputs).

With this watermark embedding method, we can now train the detector to perform transformation resilient watermark detection as follows:

$$\underset{\theta, \|\delta\|_\infty \leq \epsilon}{\mathrm{minimize}} \; \underset{s \sim P_s}{\mathrm{E}} \left[ \max_{T \in \mathcal{T}} \ell\Big( f_\theta\big(T(s + \delta)\big), 1 \Big) + \ell\Big( f_\theta\big(T(s)\big), 0 \Big) \right] \tag{1}$$

This is similar to the expectation over transformation attack (Athalye et al. (2017)), that aims to construct adversarial examples that persist through a set of transformations. During training, we only

Table 1: Transformations used to evaluate the robustness of ReSWAT in the image space.

| Transform | Transform description |
| --- | --- |
| Gaussian noise | Add Gaussian noise with standard deviation $\sigma$. |
| Rotation | Rotate an image by a random angle uniformly sampled from range $[-r, r]$. |
| Cropping | Randomly crop an image's height by $c_h$ pixels and width by $c_w$ pixels, and project back to the original image size. |
| Horizontal flip | With probability $1/2$, flip the image on the horizontal axis. |
| Brightness | Randomly adjust the brightness of an image by a factor, $b$, which ranges from 0 (no increase in brightness) to 1 (maximum whitening). |

sample from differentiable transformations, this is so we do not have to approximate the gradient, which would be prohibitively slow to training.

In practice, at each iteration in training we approximate finding $\max_{T \in \mathcal{T}}$ with $\max_{\{T_1, T_2, ..., T_n\}}$, where $T_i \in \mathcal{T}, i \in \{1, ..., n\}$. Theoretically, if we sample more transformations we achieve a better approximation for the true gradient over the full transformation distribution, however, empirically, we found it too expensive to trade more parallel transformations for longer training time. We explore how resilience to transformations is affected by the number of sampled transformations at each step in training in appendix C. All following experimental results are taken from a model trained with a sample size of one at each iteration of training.

### 3.1 OPTIMIZING FOR NON-TRANSFERABLE WATERMARKS

Traditional watermarking schemes are based on a secret key that ensures that only parties with access to the key can create watermarked content; an attacker that can watermark content without access to the key represents an integrity attack on the scheme. We refer to this kind of attack as a *specificity* attack that attempts to introduce false positives. In our case, the classifier detecting/constructing the watermark can be thought of as an approximation of a secret key.

We will now propose a method to construct watermark perturbations for a given watermark classifier, $f_\theta$, which do not transfer to other watermark classifiers trained on the same data.

Recall that given a signal $s$, to create a watermark perturbation we use PGD which takes steps in the direction of

$$-\nabla_s \max_{T \in \mathcal{T}} \left[ \ell\big(f_\theta(T(s)), 1\big) \right] \tag{2}$$

To encourage that the generated perturbation doesn't transfer to other watermark classifiers trained on the same data, we generate the watermark that is robust against an ensemble of models. Let $\{f_\theta^i : \mathcal{X} \to \{0, 1\}, i \in \{1, ..., n\}\}$ be an ensemble of watermark classifiers trained on the same problem as $f_\theta$. To limit the transferability of watermarks on $f_\theta$, we create watermark perturbations by stepping in the direction of

$$-\left( \nabla_s \max_{T \in \mathcal{T}} \left[ \ell\big(f_\theta(T(s)), 1\big) \right] + \nabla_s \max_{i, T \in \mathcal{T}} \left[ \ell\big(f_\theta^i(T(s)), 0\big) \right] \right) \tag{3}$$

Taking steps in the direction of this loss will increase the probability that watermark perturbations are only classified as watermarked by $f_\theta$ and do not transfer to other models.

## 4 EVALUATION

Here, we give experimental results that our scheme is robust to a variety of attacks. We first evaluate the scheme on image data in section 4.1 and section 4.2, and then on audio data in section 4.3.

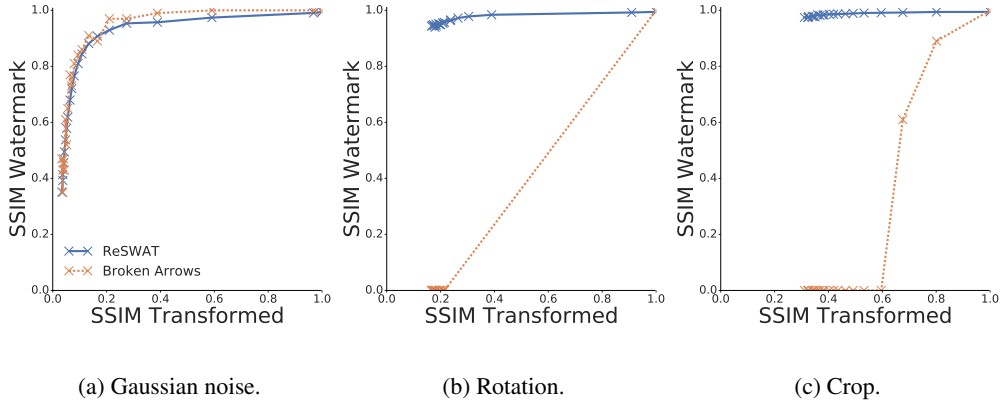

(a) Gaussian noise.    (b) Rotation.    (c) Crop.

Figure 1: Distortion needed for perfect provenance detection (by $f_{\text{comp}}^{\epsilon_{10}}$) against different signal quality degradation levels suffered by the attacker. We measure the amount of distortion the watermarking scheme must introduce in order to guarantee perfect detection accuracy under a transformation, and compare with the Broken Arrows zero-bit watermarking scheme.

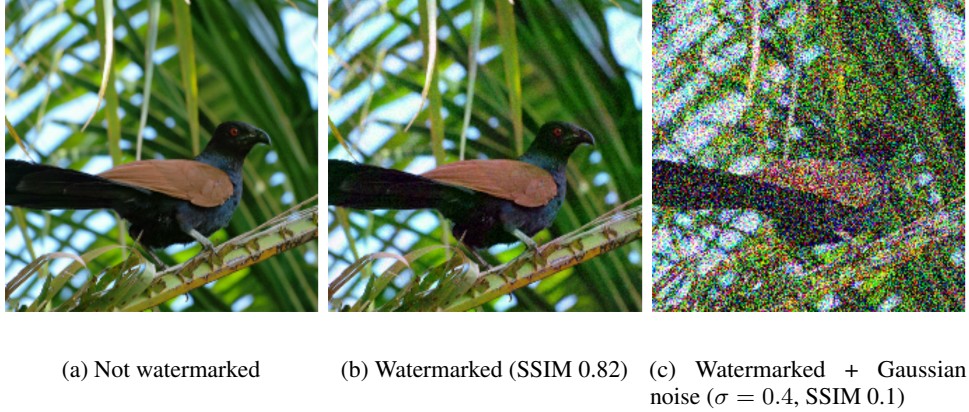

(a) Not watermarked  (b) Watermarked (SSIM 0.82) (c) Watermarked + Gaussian noise ($\sigma = 0.4$, SSIM 0.1)

Figure 2: Sample of a transformation attack on ImageNet ($f_{\text{comp}}^{\epsilon_{10}}$). (a) is a non-watermarked test set image, (b) is a watermarked version with the minimum necessary watermark required to be correctly classified by a detection model when Gaussian noise is added at a standard deviation depicted in (c).

### 4.1 EVALUATION PROTOCOL ON IMAGES

During training we sample from a set of transformations, to which the scheme should be robust. In this evaluation we use transformations detailed in table 1. We evaluate watermark detection robustness in the image space on both Cifar10 (Krizhevsky et al.) and ImageNet (Deng et al., 2009). We defer evaluation of Cifar10 to appendix A and detail only ImageNet experiments here.

**ImageNet.** We train a standard ResNet152 classifier (He et al. (2016)), modified for binary prediction, for the watermark detector. We replace batch normalization with instance normalization (Ulyanov et al. (2016)), as we noticed that batch statistics were heavily skewed by the transformations applied during training. We train with a mini-batch size of 32 for 60,000 steps, with an initial learning rate of 0.1 and decaying this by a factor of 10 every 20,000 steps.

All images are normalized into the range [0,1], and a batch of watermarked images are constructed with five PGD steps at each iteration. With respect to the set of transformations, during training, we set $\sigma$ to 0.25, $r$ to $\pi/2$, $c_h$ and $c_w$ to 10, and $b$ to 0.1. We trained four classifiers under these hyperparameters, two where at each step we randomly sample a single transformation, and watermark with $\epsilon = 5/255$ and $\epsilon = 10/255$, referred to as $f^{\epsilon_5}$ and $f^{\epsilon_{10}}$, respectively, and two where we apply a composition of all transformations, and watermark with $\epsilon = 5/255$ and $\epsilon = 10/255$, referred to as $f_{\text{comp}}^{\epsilon_5}$ and $f_{\text{comp}}^{\epsilon_{10}}$, respectively.

All watermark detection classifiers achieved $100\%$ test set accuracy, where the test set consists of 10,000 non-watermarked and 10,000 watermarked ImageNet test set images. A detailed comparison of the differences between these models is given in appendix A, however, we found that models trained under a composition of transformations are more robust to all attacks and so remaining experiments will use only $f_{\text{comp}}^{\epsilon 5}$ and $f_{\text{comp}}^{\epsilon 10}$. For evaluation, we measure the success of an attack with respect to the distortion introduced by the attack measured by structural similarity (SSIM) score (Wang et al. (2004)).

## 4.2 EVALUATING PERFORMANCE AGAINST ATTACKS

We now describe experiments measuring the quality of our watermarking scheme under various transformations seeking to induce a misclassification in the watermark detector: We study both attacks that seek to remove a watermark from given watermarked signals (thus inducing a false negative for the detector) and attacks that seek to make the detector detect a watermark in a non-watermarked image (thus inducing a false positive).

### 4.2.1 SIGNAL TRANSFORMATION ATTACKS (FALSE NEGATIVES)

We investigate the trade-off in distortion of the watermarked image when we require perfect detection under various transformations. Given a watermarked signal $s$, we sample from a space of transformations to create $n$ copies of this input under, referred to as $s^i$, $i \in \{1, ..., n\}$. We increase the watermark $\epsilon$ value until the accuracy of the detection model on $s^i$, $i \in \{1, ..., n\}$ is >99%. We measure the perceptibility of the watermark and the perceptibility of the transformation used in the attack, both in terms of SSIM, and compare with Broken Arrows (BA), where for each transformation we set the number of random samples, $n$, to 1 million.

Figure 1 shows that for various distributions of transformations the amount of distortion introduced by the watermarking scheme is dominated by the amount of distortion introduced by the transformation. For Gaussian noise, the necessary distortion introduced by the watermark (for perfect detection under a transformation) using ReSWAT is approximately equivalent to BA and substantially smaller than the distortion introduced by the transformation. While for cropping and rotation attacks, our watermarks incur negligible levels of distortion while BA fails to watermark content without incurring large distortions. We show qualitative examples in fig. 2 for a Gaussian noise attack, and analogous plots of fig. 1 using Peak-Signal-to-Noise ratio as the distortion metric in appendix B.

### 4.2.2 SPECIFICITY ATTACKS (FALSE POSITIVES)

Using the ImageNet dataset, we trained a model with the same hyperparameters as $f_{\text{comp}}^{\epsilon 5}$, however the model was optimized by constructing watermarks using eq. (3) instead of eq. (2), using five other pre-trained watermark classifiers; we denote this model by $\hat{f}_{\text{comp}}^{\epsilon 5}$. Given 20 pre-trained watermark classifiers only differing from $f_{\text{comp}}^{\epsilon 5}$ by random initialization of weights, we compare how well 1,000 watermark's constructed using each of these models transfers to $f_{\text{comp}}^{\epsilon 5}$ and $\hat{f}_{\text{comp}}^{\epsilon 5}$, representing an attack that attempts to introduce false positives. Figure 3 shows the difference in average false positive rate between $f_{\text{comp}}^{\epsilon 5}$ and $\hat{f}_{\text{comp}}^{\epsilon 5}$. At a watermark SSIM score of 0.60, $\hat{f}_{\text{comp}}^{\epsilon 5}$ has a false positive rate of below 20% while the false positive rate on these inputs is nearly 50% for $f_{\text{comp}}^{\epsilon 5}$. Clearly, optimizing eq. (3) improves resilience to specificity attacks.

### 4.2.3 CERTIFIED ROBUSTNESS

The previous sets of experiments present results on the robustness of the watermarking scheme against "best effort" attacks, i.e, we attempt to compute attacks that break the watermarking system and declare success if we fail to do so. However, it is possible that our attack algorithm failed to find the worst case transformation that would break the watermark detection. Thus, it is desirable to have a stronger guarantee against all attacks within a certain transformation space.

While provable guarantees are hard to obtain in general, we can leverage recent work on randomized smoothing techniques (Cohen et al., 2019) that are able to obtain certified guarantees against transformations constrained in the $\ell_2$ norm, i.e., the adversary can transform the signal by any amount within a distance $\epsilon$ in the $\ell_2$ norm. On 200 examples from the ImageNet test set that are watermarked

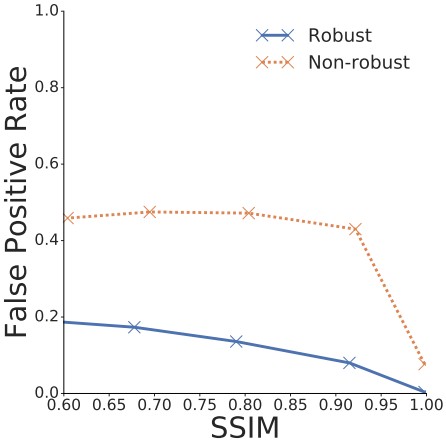
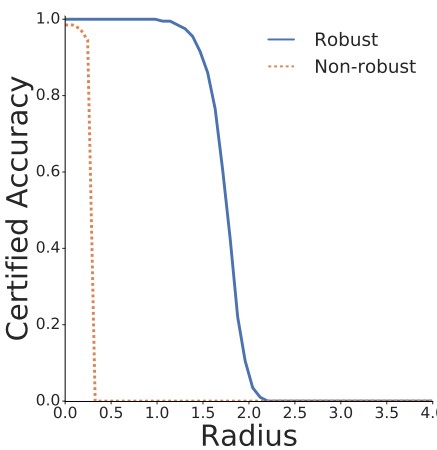

Figure 3: Average false positive rate of $f_{\text{comp}}^{\epsilon_5}$ (non-robust) and $\hat{f}_{\text{comp}}^{\epsilon_5}$ (robust) on watermark's constructed by other watermark classifiers. Given as a function of the average watermark image SSIM.

Figure 4: Certified accuracy of 200 watermarked ImageNet examples. Plotted as a function of the $\ell_2$ radius of robustness. We compare with robustness guarantees against a non-robust watermark detection model, that has not been trained on a set of transformations.

with $\epsilon = {}^{20}/_{255}$, we estimate a lower bound, $\underline{p}$, on the probability of the most-likely class under Gaussian noise parameterized by $\mathcal{N}(0, \sigma^2 I)$, using 10,000 random samples. Given $\underline{p}$, the detector is robust to adversarial perturbations, $\gamma$, if $\|\gamma\|_2 < \sigma \Phi^{-1}(\underline{p})$ (as proven in Cohen et al. (2019)), up to a confidence level of $\alpha$ which we set to 0.99. Figure 4 shows the certified accuracy of these examples as a function of the certified radius in the $\ell_2$-norm. For $\epsilon = {}^{20}/_{255}$ nearly all 200 watermarked images are robust to any perturbation with an $\ell_2$-norm smaller than 1.5. We also include results on a non-robust watermark detection model - a model trained without sampling from a distribution of transformations. This model has a comparatively small certified region of robustness, implying that training on a distribution of transformations does improve robustness.

### 4.2.4 OUT-OF-DISTRIBUTION EVALUATION

The previous section focused on adversarial attacks on signals drawn from the same distribution as the model was trained on. A natural question to ask is, would the system fail when signals are drawn from a different distribution? To evaluate this, we verified that, given a model trained on Cifar10 images, all SVHN test images are correctly classified to the non-watermarked class. Secondly, we watermarked 500 images generated by BigGAN (Brock et al., 2018) using a model trained on ImageNet samples ($f_{\text{comp}}^{\epsilon_{10}}$). We then measured resilience to transformations as detailed in section 4.2.1 – we attacked both non-watermarked and watermarked images with Gaussian noise, rotation and cropping such that the average SSIM of these images is 0.6. In comparison, the average SSIM of watermarked images was 0.82. The false positive rate (attacking non-watermarked images) was 1.2% and the false negative rate (attacking watermarked images) was 0%. BigGAN watermarked images exhibited comparative levels of resilience to transformations despite the model being trained on ImageNet samples.

### 4.3 EVALUATION ON TEXT-TO-SPEECH DATASET

To evaluate ReSWAT on audio data, we train a watermark detection model using a proprietary dataset composed of hours of high quality speech data, where each audio sample is a short speech utterance lasting between 1 and 10 seconds. We use a DeepSpeaker architecture (Li et al., 2017), modified for binary prediction. The pre-processing stage takes as input, a variable length waveform, normalizes values between -0.5 and 0.5, and outputs the fixed length mel-spectrogram which is then used as input to the model.

Table 2: Transformations used to evaluate the robustness of ReSWAT in the audio space. Transformation values specify the amount of distortion that can be added to an audio sample while maintaining perfect detection accuracy under a trained ReSWAT model. All audio samples are normalized into [-0.5, 0.5] range.

| Input | Transform | | | | |
|---|---|---|---|---|---|
| | Gaussian noise $\sigma =$ | Uniform noise $\alpha =$ | Pitch shift $\beta =$ | Silence $\gamma =$ | Roll $\delta =$ |
| Watermarked ($\epsilon = 4 \times 10^{-4}$) | 0.01 | 0.0056 | 0.16 | 0.78 | 1.0 |
| Watermarked ($\epsilon = 4.8 \times 10^{-3}$) | 0.013 | 0.0278 | 0.5 | 0.78 | 1.0 |
| | Transform description | | | | |
| Gaussian noise | Add Gaussian noise with standard deviation $\sigma$. | | | | |
| Uniform noise | Add uniform noise sampled from a range $[-\alpha, \alpha]$. | | | | |
| Pitch shift | Pitch-shifts audio by a randomly sampled scale factor in range $[-\beta, \beta]$ [1]. | | | | |
| Silence | Randomly silence a contiguous fraction, $\gamma$, of the audio. | | | | |
| Roll | Rolls values in audio by a fraction of audio length uniformly sampled at random from range $[-\delta, \delta]$. | | | | |

We train the detector model for 100,000 steps, with an exponentially decaying learning rate initialized at 0.001 and decayed by a factor of 0.9 every 1000 steps. The watermarking $\epsilon$ value is initialized at 0.04 and decreased by 0.00001 whenever the detection rate is 100% based on a moving average of the previous 100 steps. We use five PGD steps at each iteration to create watermarked inputs. During training we sample from a set of transformations, described in table 2, to which the scheme should be robust.

At test time we achieve perfect detection rate on 200 watermarked audio samples using $\epsilon = 4 \times 10^{-4}$. Similarly to section 4.2.1, we measure the robustness of watermarked audio samples with respect to the amount of distortion introduced by transformations, under the requirement that detection accuracy is >99%. For a given transformation and input, we randomly sample the input 100 times under the transformation, this creates a test set of 20,000 data points on which we measure the accuracy of the detector. Table 2 shows the results of attacking watermarked inputs for different $\epsilon$ values using various transformations. Watermarks at $\epsilon = 4 \times 10^{-4}$ are almost imperceptible, while a faint background noise can be heard for $\epsilon = 4.8 \times 10^{-3}$. The detection model is robust to a large number of transformations, for example, 78% of audio can be obscured without decreasing detection accuracy.

**Human evaluation.**

Here, we conduct human evaluation of watermark perceptibility using both mean opinion score (MOS) and A/B tests. MOS (Streijl et al., 2016) is a commonly used measure for audio quality; it is expressed as a single rational number, typically in the range 1–5,

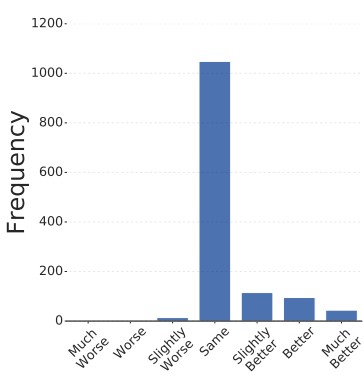

Figure 5: AB experiment. 660 non-watermarked and watermarked audio samples presented to 1320 people in total. Participants were played the original sample and watermarked sample and asked to rate how different the original audio was from the watermarked version.

where 1 is lowest perceived quality, and 5 is the highest perceived quality. The MOS is calculated as the arithmetic mean over single ratings performed by human subjects. Due to the large number of audio samples used in the study, we use a single rating per audio sample. Our dataset consisted of 2000 watermarked and non-watermarked audio samples, that were all correctly classified by the water-

mark detector model under transformations listed in table 2. The average rating of non-watermarked content was 4.595±0.576, and the average rating of watermarked content (at $\epsilon = 4 \times 10^{-4}$) was 4.530±0.530. Clearly, human participants did not perceive the watermarked audio as significantly worse or degraded.

For A/B tests we took a subset of 660 watermarked and non-watermarked audio samples. We played both the non-watermarked and watermarked audio sample to participants and asked if the watermarked audio sample was worse or better than the original. Results are shown in fig. 5; over 80% of human participants rated the quality of watermarked audio samples as the same as non-watermarked audio samples.

## 5 CONCLUSION

We presented a general solution, that leverages imperceptible watermark, to the problem of detecting the provenance of a signal. In a departure from related work, our watermarking schemes attempts to *learn* constructions of watermarks that are resilient to adversarial transformations. Our solution can be applied in numerous environments such as in image, audio and video domains. Results presented both on images and audio suggest that it is possible to construct watermarks that simultaneously maintain a high level of signal fidelity, and are resilient to adversarial transformations, with a minimal false positive rate.

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

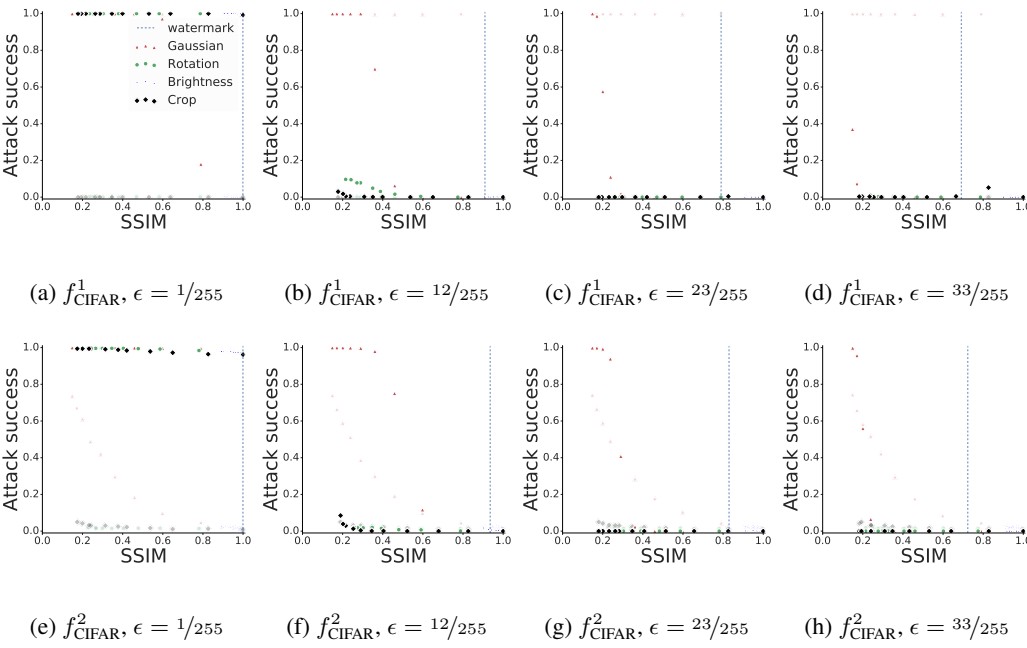

Figure 6: Transformation attack results on Cifar10. Each subfigure is plotted as function of the distortion introduced by the transformation, and we evaluate for a watermark perturbation ranging from $\epsilon = 1/255$ to $\epsilon = 33/255$. The solid markers are the attack success against watermarked images, and the faded markers, the attack success on non-watermarked images.

# A    EVALUATION ON CIFAR10 & FURTHER EVALUATION ON IMAGENET AND TEXT-TO-SPEECH DATA

**Cifar10 (Krizhevsky et al.).** We use a wide ResNet classifier (He et al. (2016)) for the watermark detector. We replace batch normalization with instance normalization (Ulyanov et al. (2016)). We trained with a mini-batch size of 32 for 60,000 steps, with an initial learning rate of 0.01 and decaying this by a factor of 10 every 20,000 steps. All images are normalized into the range [0,1]. We set the maximum size of the watermark perturbation, $\epsilon$, to $20/255$, and the watermark perturbation is constructed with five PGD steps at each iteration during training. With respect to the set of transformations, we set $\sigma$ to 0.5, $r$ to $\pi/2$, $c_h$ and $c_w$ to 2, and $b$ to 0.25. We trained two classifiers under these hyperparameters, one where at each step we randomly sample a single transformation, and another classifier where we apply a composition of all transformations, which shall be denoted by $f_{\text{CIFAR}}^1$ and $f_{\text{CIFAR}}^2$, respectively. Both classifiers achieved 100% test set accuracy, where the test set consists of 10,000 non-watermarked images and 10,000 watermarked images.

Given a transformation function $t_\theta : \mathcal{X} \to \mathcal{X}$, parameterized by $\theta$ which encodes some randomness, and an input $s \in \mathcal{X}$ with class $y \in \{0, 1\}$, we create $n$ copies of this input under $t_\theta$, $s^i$, $i \in \{1, ..., n\}$. We then measure both the SSIM of the watermarked images and SSIM of both transformed watermarked and transformed non-watermarked images as a function of the attack success over these $n$ inputs. Figure 6 shows the results of a transformation attack: for each of 2,000 Cifar10 test set images (1,000 non-watermarked and 1,000 watermarked) we create 1000 new images under a transformation; each marker in the figure is the average attack success over 1 million test images. We evaluate the attack success for a number of different watermarking $\epsilon$ values. Large $\epsilon$ increase the perceptibility of the watermark, and so correspondingly decrease the SSIM score. Note, attacking non-watermarked content is unaffected by the $\epsilon$ value chosen for watermarking.

For $\epsilon = 1/255$, the SSIM score of watermarked content is $\approx 0.99$, and thus the watermark is highly imperceptible. However, a number of attacks on watermark content succeed with high SSIM score, such as for example transformations that modify the brightness of an image. Slightly increasing the perceptibly of the watermark to $\epsilon = 12/255$, reduces the attack success of nearly all transformations to

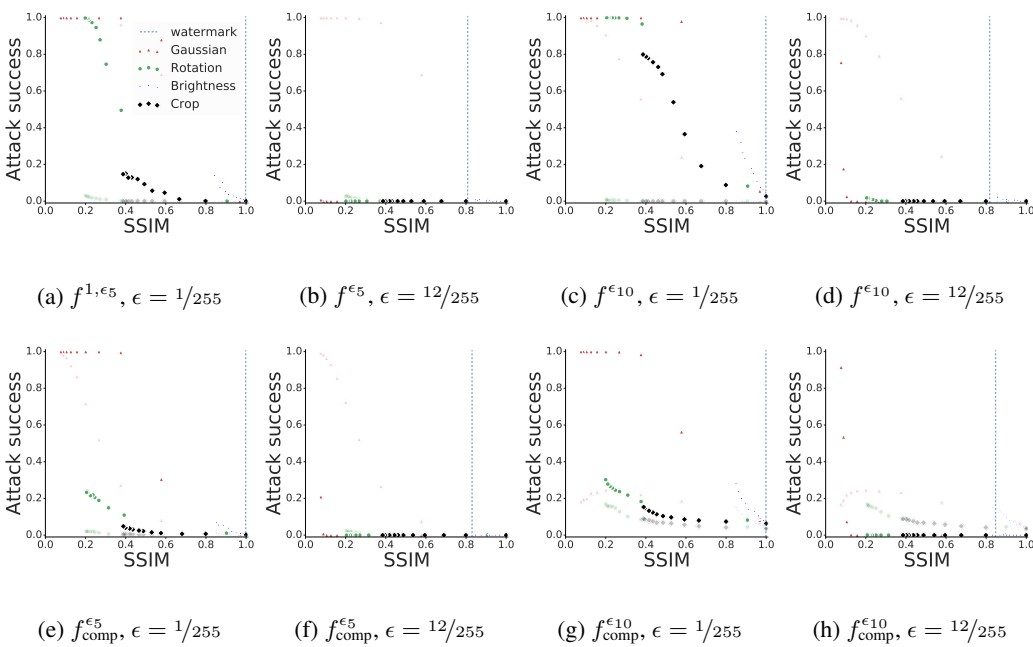

(a) $f^{1,\epsilon_5}$, $\epsilon = 1/255$  (b) $f^{\epsilon_5}$, $\epsilon = 12/255$  (c) $f^{\epsilon_{10}}$, $\epsilon = 1/255$  (d) $f^{\epsilon_{10}}$, $\epsilon = 12/255$

(e) $f^{\epsilon_5}_{\text{comp}}$, $\epsilon = 1/255$  (f) $f^{\epsilon_5}_{\text{comp}}$, $\epsilon = 12/255$  (g) $f^{\epsilon_{10}}_{\text{comp}}$, $\epsilon = 1/255$  (h) $f^{\epsilon_{10}}_{\text{comp}}$, $\epsilon = 12/255$

Figure 7: Transformation attack results on ImageNet. Each subfigure is plotted as function of the distortion introduced by the transformation, and we evaluate for a watermark perturbation ranging from $\epsilon = 1/255$ to $\epsilon = 12/255$. The solid markers are the attack success against watermarked images, and the faded markers, the attack success on non-watermarked images.

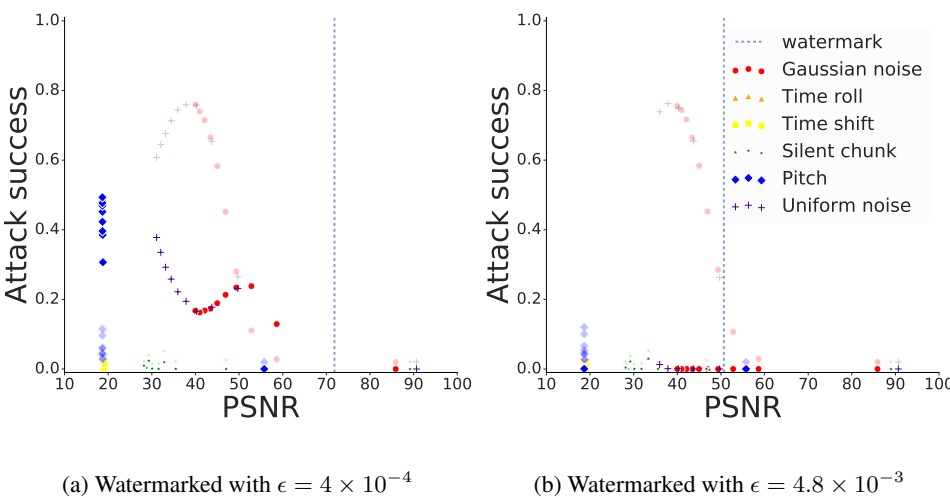

(a) Watermarked with $\epsilon = 4 \times 10^{-4}$  (b) Watermarked with $\epsilon = 4.8 \times 10^{-3}$

Figure 8: Transformation attack results on audio data. Each marker represents the average attack success versus average PSNR of 10,000 inputs sampled under a transformation. The solid markers are the attack success against watermarked images, and the faded markers, the attack success on non-watermarked images.

zero. Overall, the classifier trained on a composition of transformations is more robust to attacks on non-watermarked content.

Similar effects can be observed on ImageNet shown in fig. 7, and the propriety audio dataset in fig. 8. Interestingly, watermark classifiers on ImageNet seem to be more robust at smaller $\epsilon$ values than on Cifar10, we conjecture this is because there is a larger space in which to distribute the watermark.

## B    FURTHER COMPARISON WITH BROKEN ARROWS

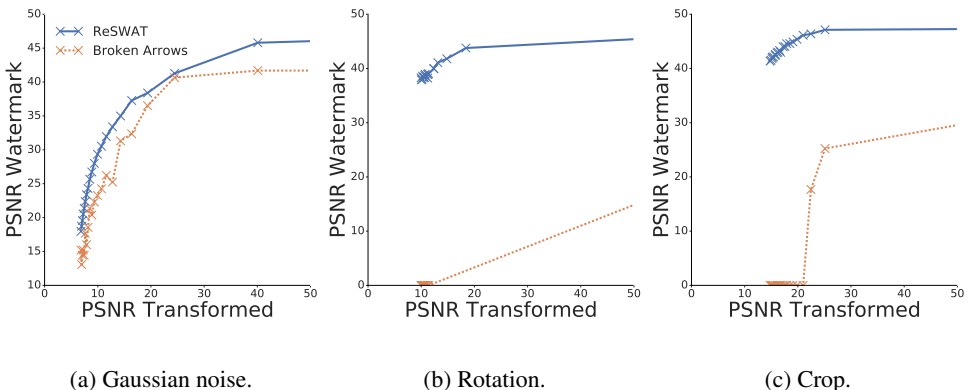

(a) Gaussian noise.          (b) Rotation.          (c) Crop.

Figure 9: Distortion needed for perfect provenance detection (by $f_{\text{comp}}^{\epsilon_{10}}$) against different signal quality degradation levels suffered by the attacker. We measure the amount of distortion the watermarking scheme must introduce in order to guarantee perfect detection accuracy under a transformation, and compare with the Broken Arrows zero-bit watermarking scheme.

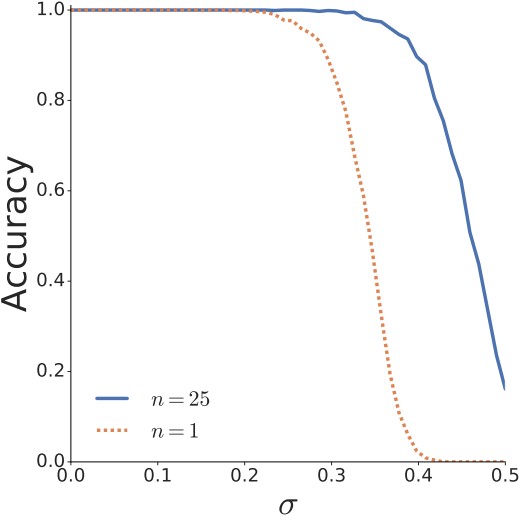

Figure 10: Comparison of ReSWAT trained using a single sample from the transformation distribution at each training step, and with 25 samples at each training step, for Cifar10, against a Gaussian noise attack.

Figure 9 gives analogous plots to fig. 1 when the measure of distortion introduced by both the watermark and a transformation is Peak-Signal-to-Noise ratio (PSNR). As one may expect, results exhibited here mirror those of fig. 1, ReSWAT and BA are comparable under Gaussian noise transformations, while ReSWAT is substantially better than BA under rotation and cropping transformations.

## C    MEASURING HOW THE NUMBER OF SAMPLES FROM THE TRANSFORMATION DISTRIBUTION USED DURING TRAINING AFFECTS RESILIENCE TO ATTACKS

Here, we evaluate the increase in resilience to transformations when we optimize ReSWAT using eq. (1) with 25 samples from the transformation distribution at each step of training, as opposed to a single sample. Figure 10 shows the resilience improvements of ReSWAT against a Gaussian noise transformation on the Cifar10 dataset. We evaluate test accuracy on 50,000 watermarked and

non-watermarked images with various levels of Gaussian noise applied. However, the improvement in resilience was at the expense of approximately a $5\times$ increase in training time.

