# OpenReview forum: "Provenance detection through learning transformation-resilient watermarking"
_ICLR.cc/2020/Conference — Reject_

### Official Review · AnonReviewer2 · 2019-10-22
**Official Blind Review #2**

**Rating:** 1

**Review:**

The paper proposes a watermark design algorithm that is based on an adversarial training paradigm where a watermark detector and a watermark generator are jointly trained to maximize the likelihood of watermark detection while bounding the manipulation of individual pixels.

I have several concerns about the clarity of the paper, its applicability to the motivating problem, and the completeness of the results that motivate my "reject" recommendation.

The description does not make it clear how the detector and watermark mechanism are jointly obtained in (1) - intuitively, a detector should know the type of watermark to be observed in order to pose a feasible detection model. The discussion focuses on "fixed detector" and "fixed watermark" cases only, but it is not clear how an iterative optimization would be initialized, and no discussion of how to set up the problem appears in the manuscript.

The motivation for the paper is also unclear. The authors discuss the need to be able to detect deep fakes, which is a relevant and meritorious problem. However, it seems that the application of the proposed approach in this setting would require deep fake creators to add watermarks to their creations, which would be a naive assumption. Watermarks are used to demonstrate provenance, but deep fake creators would not want to have such provenance verified.

The watermark transferability premise seems to rely on the random initialization of the underlying deep learning watermark detector. One could posit that if a third party is interested in creating false positives, they could train the deep learning network using a database of watermarked and non-watermarked images. Another concern here is that the formulation provided in (3) would potentially lead to low transferability for the specific classifiers provided there, but one does not know which classifier an adversary would construct - or why an adversary would construct an adversary that does not have knowledge of the specific watermark being used, or of samples of the watermark. It would have been interesting to see if it is possible to construct a good watermark detector this way (given enough training data being available). It would also be good to have some intuition as to why this robustness formulation also improves "certified robustness".

Finally, the performance comparison only considers a 10-year old watermarking algorithm; some discussion as to why this is sufficient should have been included.

Minor comments
The norm infinity subindex is missing from the minimization conditions in (1).
Figure 3 caption: watermark's -> watermarks
It is not clear why no numerical results were given for Section 4.2.4.

**Experience Assessment:**

I have read many papers in this area.

**Review Assessment: Checking Correctness Of Derivations And Theory:**

I carefully checked the derivations and theory.

**Review Assessment: Checking Correctness Of Experiments:**

I assessed the sensibility of the experiments.

**Review Assessment: Thoroughness In Paper Reading:**

I read the paper at least twice and used my best judgement in assessing the paper.

---

> ### Author Response · Authors · 2019-11-08
> **Thank you for your detailed review**
>
> We thank the reviewer for their review and detailed feedback. We will attempt to address each of the highlighted concerns.
>
> 1. Re: how the detector and watermarking mechanisms are jointly obtained: We apologize for the confusion around this. Indeed, in our scheme the watermark generation and watermark detection are jointly optimized. We only refer to a fixed watermark below Definition 2.1 as an instructive motivating example for how we can train the detector, before explaining how the joint optimization is solved in Section 3. We can think of the game of learning both a watermark and a watermark detection model as adversarial training, where the classification task is to recognise the watermark. At each step of training, given an input, x, we find a perturbation (via gradient descent) that would cause the detector model to classify this input as watermarked, we then add this perturbation to the input, x+\delta, and then update the detection model based on how strongly it classified x as not watermarked, and classified x+\delta as watermarked.
>
> 2. Re: motivation: The reviewer is entirely correct that deep fake creators would not want to add a watermark to their creations. Our threat model is designed to protect an individual or organization against abuse of data or media released by them. For example, consider an organization that has a corpus of photos/audio samples/videos, or a generative model that they allow the public access to query. These samples could then be used to create deep fakes by another party. If the organization has watermarked these samples with our scheme, the watermark is designed to be robust to post-processing transformations that an adversary might introduce, and so the watermark will persist when the deep fake creator manipulates the sample. The organization can then verify if the deep fake was originally created using samples that they own.
>
> 3. Re: transferability: An attacker that has access to content that contains both non-watermarked and watermarked versions, is currently outside of our threat model. As far as we can tell, this is a relatively uncommon assumption in zero-bit watermarking, as if an attacker has a procedure to identify if a piece of content was or was not watermarked they have practically already broken the scheme. Furthermore, to increase the cost of this detection attack, in potential practical applications an organization can limit the exposure to watermarked content by prohibiting any single party from making a large number of queries to access content. Our experiments in Section 4.2.2 are designed to show that an attacker that does not have this information, but has other information about the watermarking procedure, cannot break the scheme. The attacker has full knowledge of the architecture of the detector model, the algorithm that is used to create watermarks and the detector, hyperparameters used in the algorithm, and the exact training set used, and so represents a relatively strong attack.
>
> 4. Re: certified robustness: Section 4.2.3 attempts to show that a detector model that has been trained with post-processing transformations (eq. 1) is more robust than a model trained without post-processing transformations (eq. 1 with max over T omitted from the formulation). The certification procedure provides a certified region of robustness to Gaussian noise (correspondingly an L2 norm ball). Since Gaussian noise is included in the set of post-processing transformations, the model that has been trained to be robust to these transformations is naturally more robust to Gaussian noise (and so has a larger certified area with respect to the L2 norm) than the model that has not been trained on these set of transformations.
>
> 5. Re: Validity of comparing with Broken Arrows: Please refer to our response to Blind Reviewer 1 regarding the concern of the comparison with Broken Arrows.
>
> 6. Re: No numerical results in Section 4.2.4: The results on the SVHN test set is a single number, reporting that no mistakes were made by the detector. Results on BigGAN also had a false positive and false negative rate of <1.2% when attacked with various post-processing transformations. We have updated the paper now detailing these results.
>
> Thank you again for the helpful review, we would be more than happy to act on any specific recommendations that the reviewer would consider important to revising the score for this paper.

---

### Official Review · AnonReviewer1 · 2019-10-23
**Official Blind Review #1**

**Rating:** 6

**Review:**

The authors introduce ReSWAT, a method for transformation-resilient watermarking of images via adversarial training. The high level idea is to learn a watermark/detector pair (W,D). W can be any transformation (in this paper, an l-infty bounded perturbation) that imputes an imperceptible distortion to a given input, while D is a detector that distinguishes watermarked from non-watermarked images. There is an additional requirement that the detector should be robust to simple transformations such as rotations, cropping, flipping, and contrast enhancement.

The authors pose a min-max style learning problem for learning (W,D) that leads to a natural adversarial training scheme. In particular, it can be viewed as creating an adversarial defense to the Expectations-over-transformations attack of Athalye et al. Experimental results on a bunch of different datasets confirm that the method works,

The paper is well-written and the contributions are clear. In terms of conceptual or theoretical novelty, the paper is limited (the method essentially boils down to regular adversarial training with a slightly non-standard loss function) but the connection to watermarking seems novel and is nicely executed.

My only concern is the lack of comparisons with baselines. I am not a digital forensics expert -- so I don't know what the state of the art is -- but the only comparisons made are with the Broken Arrows (BA) watermarking , which seems to be over 10 years old, so I am not sure how to evaluate the results.

**Experience Assessment:**

I have read many papers in this area.

**Review Assessment: Checking Correctness Of Derivations And Theory:**

N/A

**Review Assessment: Checking Correctness Of Experiments:**

I assessed the sensibility of the experiments.

**Review Assessment: Thoroughness In Paper Reading:**

I read the paper at least twice and used my best judgement in assessing the paper.

---

> ### Author Response · Authors · 2019-11-08
> **Thank you for your comments**
>
> We thank the reviewer for their careful review and thoughtful feedback.
>
> A concern was raised in the reviews that since Broken Arrows was proposed ten years ago, it might not reflect the state of the art. We had performed an extensive literature survey of zero-bit watermarking algorithms during our research. We used Broken Arrows to compare against our method for the following reasons:
> 1. To the best of our knowledge, no publicly available zero-bit watermarking scheme has been published that claims superiority over Broken Arrows and is still used as the state-of-the-art watermarking method by other works. For example, Quiring et al. (2018) [1] used Broken Arrows in their recent work (see Section 4.1 in [1]).
> 2. Broken Arrows won an international competition designed for defending against watermarking attacks [2].
> 3. It has provable robustness to Gaussian noise attacks (see Furon & Bas (2008) [3]).
>
> [1] Quiring, Erwin, Daniel Arp, and Konrad Rieck. "Forgotten siblings: Unifying attacks on machine learning and digital watermarking." 2018 IEEE European Symposium on Security and Privacy (EuroS&P). IEEE, 2018.
> [2] http://bows2.ec-lille.fr/
> [3] Furon, Teddy, and Patrick Bas. "Broken arrows." EURASIP Journal on Information Security 2008.1 (2008): 597040.

---

### Official Review · AnonReviewer3 · 2019-10-27
**Official Blind Review #3**

**Rating:** 8

**Review:**

This paper is about a novel method to add watermarks to images and audio that is highly robust to several transformations that is closely related to gan methods. The idea is that the watermark signal is learned concurrently to the detector network, which share similarities to a generator and detector networks. Five standard attack transformations are considered and a specific optimization to reduce the transferability of the watermark is considered. The method is compared against Broken arrows on Cifar10 and Imagenet. It shows similar or better performance for Gaussian noise attack given the same amount of perturbance allowed in a signal while very much better performance for the other attacks. Going beyond the five attacks, the paper also includes an estimation of the probability of confidence of finding watermarked images given a fixed l2 norm radius. Finally the method is also tested on audio on a proprietary dataset with a deepspeaker architecture which still shows very good performance and it is confirmed by a human evaluation where participants found the watermarked audio not significantly worse or degraded.

The paper is well written and of excellent presentation. The proposed method is novel and goes in the interesting direction of learning watermarks using adversarial training techniques. Experiments shows that the method has good performance but it is only compared to Broken Arrows (i.e. a zero bit watermarking) which is the state of the art of this type of watermarking. It would have been interesting a comparison to other key based watermarking methods to also deeply evaluate the signal transformation attacks. All in all, I found the paper to be significant and I would like to see it accepted.

**Experience Assessment:**

I have read many papers in this area.

**Review Assessment: Checking Correctness Of Derivations And Theory:**

I assessed the sensibility of the derivations and theory.

**Review Assessment: Checking Correctness Of Experiments:**

I assessed the sensibility of the experiments.

**Review Assessment: Thoroughness In Paper Reading:**

I read the paper at least twice and used my best judgement in assessing the paper.

---

> ### Author Response · Authors · 2019-11-08
> **Thank you for your comments**
>
> We thank the reviewer for their careful review and thoughtful feedback. We agree that an exciting direction for future work would be to compare this scheme to other key-based watermarking schemes. In this work, we decided to focus on zero-bit watermarking as it most naturally aligns with the goal of provenance detection. However, it is true that the scheme could be extended in the future to multi-bit watermarking schemes and in that case it would be natural to compare against other key-based multi-bit watermarking schemes. Please refer to our response to Blind Reviewer 1 regarding the concern of the comparison with Broken Arrows.

---

### Decision · Program_Chairs · 2019-12-19

**Decision:**

Reject

**Comment:**

This paper offers an interesting and potentially useful approach to robust watermarking.  The reviewers are divided on the significance of the method.  The most senior and experienced reviewer was the most negative.  On balance, my assessment of this paper is borderline; given the number of more highly ranked papers in my pile, that means I have to assign "reject".